# Interface Control in Additive Manufacturing of Dissimilar Metals Forming Intermetallic Compounds—Fe-Ti as a Model System

**DOI:** 10.3390/ma13214747

**Published:** 2020-10-23

**Authors:** Di Cui, Antaryami Mohanta, Marc Leparoux

**Affiliations:** Empa–Swiss Federal Laboratories for Materials Science and Technology, Laboratory for Advanced Materials Processing, Feuerwerkerstrasse 39, CH–3602 Thun, Switzerland; di.cui@empa.ch (D.C.); antaryami.mohanta@empa.ch (A.M.)

**Keywords:** laser metal deposition, dissimilar metals, intermetallic compounds, chemical composition, microhardness

## Abstract

Laser metal deposition (LMD) has demonstrated its ability to produce complex parts and to adjust material composition within a single workpiece. It is also a suitable additive manufacturing (AM) technology for building up dissimilar metals directly. However, brittle intermetallic compounds (IMCs) are formed at the interface of the dissimilar metals fabricated by LMD. Such brittle phases often lead to material failure due to thermal expansion coefficient mismatch, thermal stress, etc. In this work, we studied a Fe-Ti system with two brittle phases, such as FeTi and Fe_2_Ti, as a model system. Fe was grown on top of Ti at various process parameters. The morphologies and microstructures were characterized by optical microscopy (OM) and scanning electron microscopy (SEM). No cracks along the interface between pure Ti and bottom of the solidified melt pool were observed in the cross-sectional images. Chemical composition in the fabricated parts was measured by Energy-dispersive X-ray spectroscopy (EDS). Electron backscatter diffraction (EBSD) was performed in addition to EDS to identify the crystalline phases. The Vickers hardness test was conducted in areas with different phases. The chemical composition in the melt pool region was found to be a determining factor for the occurrence of major cracks.

## 1. Introduction

Laser metal deposition (LMD) is an advanced powder-injective laser additive manufacturing (AM) technology, which is capable of directly producing dense metal parts with complex geometry and especially with varying composition. The ability of LMD has been demonstrated in the fields of rapid manufacturing, repairing, and remanufacturing of metallic components [1]. Thanks to convenient switching of powder feedstock during the deposition process, LMD has a unique advantage in fabricating a workpiece of dissimilar metals.

Titanium alloys exhibit a high strength-to-weight ratio, biocompatibility, and superior heat and corrosion resistance. Therefore, they are considered as excellent engineering materials in biomedical, aerospace, automobile, nuclear, and many other industries [1]. Stainless steel (SS) has been widely used in the fields of automobile, petrochemical, construction, power generation, and medical devices due to its excellent mechanical properties, corrosion resistance, good weldability, and low cost [2]. In order to combine good mechanical and metallurgical properties with a more affordable cost, there is increasing interest in joining Ti (-alloy) and SS [3,4,5,6].

Various techniques, including LMD, have been employed to join Ti (-alloy) and SS [1,4,7,8]. All have been confronted with cracking issues due to the presence of FeTi and Fe_2_Ti brittle intermetallic compounds (IMCs) at the interface. A Fe-Ti phase diagram showing the different phases that may form at equilibrium is presented in Figure 1 [9].

Chen et al. [4] used direct laser welding to join Ti and SS304 with the laser beam centered on the contact line. The ultimate tensile strength (UTS) of the joint was only 65 MPa, and continuous distribution of large amounts of Ti-Fe IMCs was found at the interface. Offsetting the laser beam by 0.6 mm toward SS304 increased the UTS to 150 MPa. Smaller and discontinuous Ti-Fe IMC zones were formed in the weld thanks to the reduction of melted Ti. Zhang et al. [8] achieved a UTS of 336 MPa in a joint between Ti6Al4V and SS301L by offsetting the laser beam by 0.45 mm toward Ti6Al4V. They inferred that the temperature at the contact line was below the melting point of Ti6Al4V and above that of the β-Ti + FeTi eutectic. Atomic interdiffusion promoted by the elevated temperature created eutectic liquid locally at the contact line, which later expanded. Fe and Ti in the liquid further diffused into Ti6Al4V and SS301L, respectively, to form a β-Ti(Fe) solid solution and Fe_2_Ti IMC. The UTS increased thanks to the finer eutectic structure. Li et al. [1] tried to directly deposit SS316 on Ti6Al4V using LMD with a 2 mm wide laser beam. The deposited layer fell off from the substrate with a clear cracking sound. The researchers then inferred that it is impossible to join the two directly with their setup, and proceeded with a transition route with three filler materials (Ti6Al4V-V-Cr-Fe-SS316) to avoid the formation of Ti-Fe IMCs. A smooth gradient in hardness and composition was achieved from Ti6Al4V through interlayers to SS316. This approach provided a solution for crack-free joining under certain circumstances. However, the non-biocompatible elements V and Cr used as adaptation layers limited the application of this solution in the medical industry [8].

Despite the interest of building parts out of Ti and stainless steel, no direct joining using an additive manufacturing approach has been reported to the best of the authors’ knowledge. Most of the elements of the stainless steel may form intermetallic compounds among them, like σFeCr, or by reaction with Titanium like αCr_2_Ti, βCr_2_Ti, γCr_2_Ti, and Ti_5_Cr_8_Fe_16_ [10]. In this fundamental study aiming at understanding the formation of the different phases at the interfacial zone during a laser deposition process, the material system has been simplified to Ti and Fe. Iron is indeed the main constituent of SS and both the FeTi and Fe_2_Ti IMCs, reported as main critical brittle phases during the joining of Ti(-alloy) and SS, may also form during the laser processing. The morphology, microstructure, chemical composition, and microhardness of the material and the interfacial zone were investigated as a function of the main AM processing parameters.

## 2. Materials and Methods

### 2.1. Materials

Ti Grade 1 plates with a thickness of 4 mm (Zapp AG, Ratingen, Germany) were used as substrate for depositing Fe. Non-spherical Fe micropowder (Shanghai Knowhow Powder-tech Co., Ltd., Shanghai, China) sieved for 45–125 µm was used as feedstock for the deposition. According to the provider’s certificate, Fe purity is 99.5 wt.%.

### 2.2. Deposition of Fe on Ti

A commercial LMD machine (Mobile 1.0, BeAM, Strasbourg, France) was used for building the 3D structures in the shape of single tracks and single walls.

An airtight process chamber of the LMD system offered the possibility to work under a controlled atmosphere. A continuous wave (CW) fiber laser with a maximum power of 500 W, operating at a wavelength of 1068 nm (YLR-Series, IPG Photonics), was used as the energy source. The laser has a Gaussian transverse profile and the focal spot diameter of the laser is about 800 µm. A volumetric powder feeder (Medicoat AG, Mägenwil, Switzerland) was equipped with two powder containers. The interior of the process chamber and the powder feeder with two powder containers are shown in Figure 2.

The Fe microscale powder was fed by the volumetric conventional powder feeder and injected coaxially to the laser spot in the process zone through a specific conical nozzle. The focus distance of both the laser beam and the nozzle for the powder jet was 3.5 mm. The nozzle was supported by a 3 axes system (x, y, z) and the substrate was fixed on a 2 rotative-axes holder (a, b). The samples were produced in argon atmosphere (40 ppm O_2_, 200 ppm H_2_O).

The key parameters of the deposition process are scan speed (S), laser power (P), and powder feed rate (F). The ratio P/S characterizes the linear energy (LE). Additionally, for wall samples, NL is the final number of layers deposited on the same position. Each spot on the line was irradiated once during the deposition of each layer of the wall, and the interval between two subsequent irradiations is the cooling time (CT).

All single tracks and walls were designed to be 10 mm long. During printing of the walls, the nozzle always moved up by 0.2 mm to start a subsequent layer. Speed, power, and feed rate were studied in groups Line_S, Line_P, Line_LE, and Line_F, respectively. In group Wall_NL, NL increased from 1 to 5 with a CT of 3 s for all walls. In group Wall_CT, CT decreased from 8 to 1 s with an NL of 3 for all walls. The parameters for all samples are summarized in Table 1. The combination of power of 150 W, speed of 1000 mm/min, and feed rate of 4.8 g/min is referred to as the standard parameter. Single tracks and walls deposited with the standard parameters are marked light and dark grey, respectively, in Table 1.

### 2.3. Characterization Methods

The produced samples were cut perpendicular to the laser scanning directions into halves. Each sample then had two parts. The cross-sections were then embedded in Demotec 4000 resin, ground up to 2500 grit with SiC grinding paper, and polished with 6 and 3 μm diamond pastes, and finally, with an OPS solution (0.04 μm SiO_2_ with 10% H_2_O_2_).

Cross-sectional morphology and microstructure were examined with an optical microscope (OM) (ZEISS Axioplan, Munich, Germany) and a scanning electron microscope (SEM—Mira Tescan 3, Brno, Czech Republic). An energy-dispersive X-ray spectrometer (EDS—Ametek Edax Octane plus, Mahwah, NJ, USA) was used to investigate the elemental spatial distribution. An electron backscatter diffraction camera (EBSD—Ametek, Mahwah, NJ, USA) was used to investigate the crystalline structure. The Vickers hardness was measured with a load of 10 or 100 g and a dwell time of 10 s (ESI Prüftechnick GmbH, LMA-302-VLX, Wendlingen, Germany).

## 3. Results and Discussion

As seen in Table 1, one standard processing condition set was repeated at least five times for assessing the reproducibility of the LMD process. Therefore, two geometrical characteristics, such as deposition height and penetration depth, were measured for each cross-section. The deposition height (H) is the height of the solidified melt pool above the surface of the substrate and has an average of 67 μm, a standard deviation of 13 μm, and a coefficient of variation of 19.5%. The penetration depth (D) is the depth of the solidified melt pool below the surface of the substrate and has an average of 35 μm, a standard deviation of 3 μm, and a coefficient of variation of 7.7%. H + D corresponds to the total height of the deposited material including the mixing with the substrate material. In the following, a general error on the geometrical measurements for H and D is defined based on these coefficients of variation for all the experiments.

### 3.1. Group Line_S

Single line samples (group Line_S) were fabricated with increasing scan speed at a constant laser power and powder feed rate. OM images of cross-sections and measured geometries, i.e., the penetration depth D into the substrate and the height of the deposited layer H, of these samples are shown in Figure 3. 

A spherical pore with a diameter of around 50 μm was observed in Line_S_1. One unmelted particle with a diameter around 70 μm observed in Line_S_2 was identified as Nb contamination from previous experiments. In this experimental series, neither cracks nor delamination at the interface between the substrate and the deposited track could be observed, even though small and discontinuous cracks have been found in the deposited layer. Some long straight lines crossing the substrate and the layer might still be observed on the cross-sections as, for instance, in Figure 3(Line_S_1). These lines were produced by the polishing process. 

As the scan speed increased, both H and D of the single track decreased. H decreased from 164 ± 33 to 40 ± 8 μm, whereas D decreased from 56 ± 4 to 17 ± 1 μm. A brown thin band with varying thickness appeared at the bottom of each track. EDS line scans have been done across the interface and the deposition to obtain atomic concentrations of Ti and Fe. All scan lines were vertical, starting from pure Ti slightly below the interface to the top of the deposition. The EDS line scan across the sample Line_S_1 is shown as an example in Figure 4 on the right. The scale of the line scan has been adapted to the layer thickness of the corresponding sample shown in Figure 4 on the left.

The composition changes sharply from Ti in the substrate to Fe_30_Ti_70_ (at.%) in the brown band corresponding to the β-Ti + FeTi eutectic in the phase diagram. On top of the Ti-rich eutectic band, a discontinuous yellowish phase was observed with a composition corresponding to Fe_50_Ti_50_ by EDS point analysis. Further up to the top of the deposited track, the concentration changes to Fe_90_Ti_10_, corresponding to the solubility limit of Ti in α-Fe, and fluctuates around that. The upper part of all tracks exhibits a similar light grey color.

### 3.2. Group Line_P

No material was deposited for P ≤100 W (samples Line_P1 and P2). Thus, only the samples deposited at higher powers are presented in Figure 5. A spherical pore with a diameter of around 60 μm was observed in Line_P_3. No crack or delamination at the interface was observed for all the samples. A crack crossing the entire deposited track was found in Line_P_4 and Line_P_5. No crack penetrated into the Ti substrate.

As the laser power increased, D predominantly increased from 30 ± 2 to 280 ± 20 μm, whereas the influence of the power on H was not significant. The same Ti-rich eutectic band was found in all three samples, and this band in Line_P_5 is obviously wider. The optically visible features and chemical composition profile of Line_P_3 are in accordance with Line_S_3, which was fabricated with the standard parameters. As P increased, the amount of yellowish phase increased. In Line_P_4, the composition changes from Ti to Fe_30_Ti_70_ in the brown band, then fluctuates between Fe_30_Ti_70_ and Fe_50_Ti_50_, and finally, between Fe_50_Ti_50_ and Fe_60_Ti_40_. In Line_P_5, the composition changes from Ti to Fe_30_Ti_70_ in the brown band, then fluctuates between Fe_30_Ti_70_ and Fe_50_Ti_50_, and finally, between Fe_50_Ti_50_ and Fe_40_Ti_60_.

### 3.3. Group Line_LE

The results in the Line_LE samples produced with a constant P/S but variation of P and S were very similar to that of Line_P samples. OM images of the cross-sections with atomic concentration overlay and measured geometries are shown in Figure 6. Spherical pores with diameters of around 10 μm were observed in Line_LE_3. Pores with diameters of around 50 μm were observed in both Line_LE_3 and Line_LE_5. No cracks or delamination at the interface were observed for all the samples. A crack crossing almost the entire deposited track was found in Line_LE_4 and Line_LE_5. No crack penetrated into the Ti substrate.

As P and S increased proportionally, H increased slightly from 80 ± 16 to 99 ± 20 μm. D increased, however, significantly from 32 ± 2 to 111 ± 9 μm, and finally, to 175 ± 13 μm. The optically visible features and chemical composition profile of Line_P_3 are in accordance with standard samples, confirming the reproducibility of the process. In Line_LE_4, the composition changes from Ti to Fe_30_Ti_70_ in the brown band, then fluctuates around Fe_60_Ti_40_ up to the top of the layer. In Line_LE_5, the composition changes from Ti to Fe_30_Ti_70_ in the brown band, then fluctuates between Fe_30_Ti_70_ and Fe_50_Ti_50_.

### 3.4. Group Line_F

Samples in group Line_F have been fabricated with an increasing powder feed rate at a constant laser power and scan speed. No cracks or delamination at the interface were observed for all the samples. As F increased from 2.4 to 6.8 g/min, H decreased from 86 ± 17 to 52 ± 10 μm, and D fluctuated around 30 μm with no obvious trend. The composition transitions are similar for all samples and comparable to the changes observed in the standard samples. The composition in the upper part of Line_F_5 is Fe_66_Ti_33_, indicating the presence of a Fe_2_Ti IMC for the highest investigated feed rate of 6.8 g/min.

### 3.5. Group Wall_NL

After single track experiments, multilayer walls have been produced using the standard parameters. The number of layers has been increased from 1 to 5 and a constant cooling time of 3 s between two subsequent depositions has been applied. OM pictures of the cross-sections with measured geometries are shown in Figure 7. A few small cracks were found in Wall_NL_3 and Wall_NL_4 through the Ti-rich eutectic band and a few large spherical pores are visible. No cracks or delamination through the interface were observed for all samples. 

The optical microscope pictures did not allow distinction of the interface between the two subsequent layers. However, H increased almost linearly with the number of layers from 60 ± 12 to 450 ± 90 μm, showing an average increase of around 100 μm per additional layer. D slightly increased from 30 ± 2 μm for one layer (Wall_NL_1) to 40 ± 3 μm for the two-layer sample (Wall_NL_2). The deposition of additional layers does not seem to increase the penetration depth, which remains around 40 μm. Composition in Wall_NL_1 reached Fe_80_Ti_20_ within the first 50 μm and this composition remained until the top of the track. For the other samples, the compositions all reached Fe_90_Ti_10_ in the first 100 μm, and then, gradually shifted to almost pure Fe.

### 3.6. Group Wall_CT

Considering the standard deposition parameters, the cooling time between two deposition steps was varied from 8 to 1 s. Only three layers were deposited. Similar to the previous samples, small cracks and voids were observed, although no delamination or cracks were observed along the interface. The cooling time seems to have no influence on the penetration depth and only a slight increase in height is observed for CT <3 s. Additionally, the composition transitions are similar to the one measured in Wall_NL samples. Therefore, the cross-sections of these samples were not depicted here.

### 3.7. Microhardness, EDS, and EBSD Results from Line_P_5 and Wall_NL_5

To assess the hardness of the different IMCs as well as the pure metals, indents were performed on some samples. The corresponding local Vickers microhardness values are presented in Figure 8 for the samples Line_P_5 and Wall_NL_5.

In Line_P_5, 736 ± 46 HV_0.01_ was measured in the Ti-rich eutectic band and 598 ± 82 HV_0.01_ in the main body of the track with an estimated composition fluctuating between Fe_30_Ti_70_ and Fe_50_Ti_50_.

In Wall_NL_5, microhardness is 652 ± 1 HV_0.01_ in the Ti-rich eutectic band. A jump to as high as 1052 HV_0.01_ was found slightly above the band, where the average microhardness is 967 ± 85 HV_0.01_. Then, the microhardness decreases gradually to 178 ± 10 HV_0.01_ in the wall body.

Multipoint EDS measurements have been done in the bottom left area in Figure 8b around the highest measured microhardness, i.e., 1052 HV_0.01_. Corresponding microhardness and possible phases are shown in Table 2. Crystal structures are taken from the Springer Materials database [11].

A SEM image taken at the upper border of the Ti-rich eutectic band in Wall_NL_5 is shown in Figure 9a. EBSD was performed in a slightly larger area. An image quality (IQ) map obtained from EBSD is shown in Figure 9b, whereas the phase map with the Confidence Index (CI) is shown in Figure 9c. As phases with the same atom arrangement and similar lattice parameters cannot be distinguished with EBSD, b.c.c. generic and h.c.p. generic were used for indexing. Morphologies observed in the SEM image (Figure 9a) are also found in the IQ map (Figure 9b). The Fe_30_Ti_70_ area according to EDS measurements was indexed as b.c.c. with relatively low CI, corresponding to the eutectic between β-Ti (b.c.c.) and FeTi (b.c.c.). The Fe_50_Ti_50_ area was indexed as b.c.c. with the same structure as FeTi. The thin Fe_73_Ti_27_ band was indexed as h.c.p. with relatively low CI and could corresponded to Fe_2_Ti. In the Fe_84_Ti_16_ area, two lamella structures with thickness of 100~300 nm were observed. The lamella structure with higher IQ (brighter in the IQ map) was indexed as b.c.c. with high CI, corresponding to α-Fe. The other lamella structure with lower IQ was indexed as h.c.p. with low CI, corresponding to Fe_2_Ti. The transition through FeTi and Fe_2_Ti IMCs is limited within a small thickness of 4 μm in the investigated area.

### 3.8. Discussion

Within the investigated process parameters window and despite the possible formation of brittle IMCs in the Fe-Ti material system, neither delamination nor cracks could be observed along the interfacial zone between these two metals. Only a few large pores and small cracks within the layers could be seen by optical microscopy. However, these pores and cracks could not be correlated to specific energy inputs. This good interfacial bonding appears in contradiction with the extensive delamination reported by Li et al. [1], while investigating the LMD processing of the Ti-SS material system. These authors used, however, large laser diameter and power, 2 mm and 1000 W, respectively, which ensured melting a large amount of Ti. Their scan speed and powder feed rate were also relatively low, i.e., 200 mm/min and 5.1 g/min, respectively, which indicates the melting of a large amount of SS316. It can be inferred that a large melt pool had formed with an efficient mixing of both Fe and Ti promoting the formation of brittle IMCs.

The additive manufacturing process requires a metallurgical bonding between the deposited layers for building 3D parts. It is then necessary to melt the support layer and to achieve an overlapping zone between the two subsequent layers. However, as described above, an enrichment of the melt pool by both metals will promote the formation of the IMCs. Therefore, the penetration depth into the bottom layer has to be controlled carefully. Wolff proposed a qualification of the metallurgical bonding by introducing a dimensionless parameter called dilution (*d*), being [12]:(1)$$d=\;\frac{D}{H+D}$$
where *D* is the penetration depth and *H* is the height of the material deposited above the substrate as described previously in this study. The optimum, according to Wolff, should be a dilution between 10% and 30%. In the present investigation, this dilution varies between almost 20% and more than 60%. The high dilution values are obtained for high-energy inputs, especially for the highest powers investigated. Indeed, it seems that power is the most critical process parameter influencing the penetration depth. At high powers, a large amount of Ti coming from the substrate is available for building IMCs. For the highest power investigated here, i.e., 300 W, the composition up to the top of the single track is even richer in Ti than in Fe (Fe_40_Ti_60_). The tracks deposited with high powers show one or even many cracks, starting from the interface and propagating straight up to the top of the deposited layer. However, no cracks along the interface between the Ti substrate and the iron deposition could be seen. The brittle IMCs FeTi and Fe_2_Ti have been identified based on compositional EDS measurements and their respective crystalline structures could be confirmed by EBSD analyses. These phases are always observed above a first Ti-rich phase present along the interface and having a similar or even higher hardness as compared to the FeTi phase.

Although laser power is the dominant factor influencing the penetration depth, the highest heights of single tracks are obtained for lower scan speeds. No correlation could be drawn regarding the formation of large pores found in a few samples. As these pores are spherical, they may form due to gas trapping during the printing process. Further investigations are then required to avoid their formation by adjusting the shielding gas flow rate used to protect the optical setup of the laser head as well as the carrier gas flow to transport the metal powders.

To achieve 3D structures, more than one layer is normally required. The deposition of additional Fe layers leads to a partial remelting of the sub-layer down to the interfacial zone. Indeed, under the standard processing conditions used in this study, the penetration depth slightly increases from 30 to 40 µm after the second deposition step. This penetration does not change anymore with the application of the additional layers. The height seems to increase linearly with the number of layers, even if the first layer in contact with the substrate may be slightly thinner, i.e., 60 µm instead of 100 µm under the standard deposition conditions. It was not possible to distinguish the successive layers on the cross-sections to assess this linear increase. The cooling time variations indicate that already after 1 s, the sublayers are probably completely solidified. For all the produced multilayered walls, the composition changes gradually from Ti in the substrate to almost pure Fe within a thin transition zone. Despite the high cooling rates making additive manufacturing a non-equilibrium process, all the specific compositions of the phase diagram could be identified as seen in Table 2. However, the defined processing parameters for building 3D parts allow the confinement of the brittle IMCs FeTi and Fe_2_Ti within a thickness of only 4 µm, which could avoid cracking. The remelting of part of the first layer leads to the dilution of titanium in iron and the composition moves from Fe_80_Ti_20_ to Fe_90_Ti_10_ within the first 100 µm before changing to pure Fe. Above this, the composition is close to pure Fe, where Ti is no longer present.

## 4. Conclusions

This work shows the capability of direct deposition of Fe walls on Ti substrate. Laser power and scan speed have been found to be dominant in determining the amount of Ti and Fe in the melt pool, respectively. The investigated parameters induced a sharp and gradual composition change at the interface with the Ti substrate. All the crystalline phases of the binary phase diagram could be found at the interface thanks to EDS and EBSD characterizations, despite the non-equilibrium state of the laser process, however with different thicknesses depending mainly on the laser power driving the penetration depth. The brittle IMCs are thus confined within a thin zone that may avoid extensive crack formation and propagation. The deposition of subsequent layers does not influence the interfacial zone anymore, indicating that the printing parameters could then be optimized for depositing the second metal. This study demonstrates then that the first 2–3-layer deposition tailors the interface between two dissimilar metals where cracks and delamination would occur. It is worth noting that for larger dimensions of the printed parts, the stresses may be more important. This will be investigated in a future study aiming at also evaluating the mechanical performances of the printed structures.

## Figures and Tables

**Figure 1 materials-13-04747-f001:**
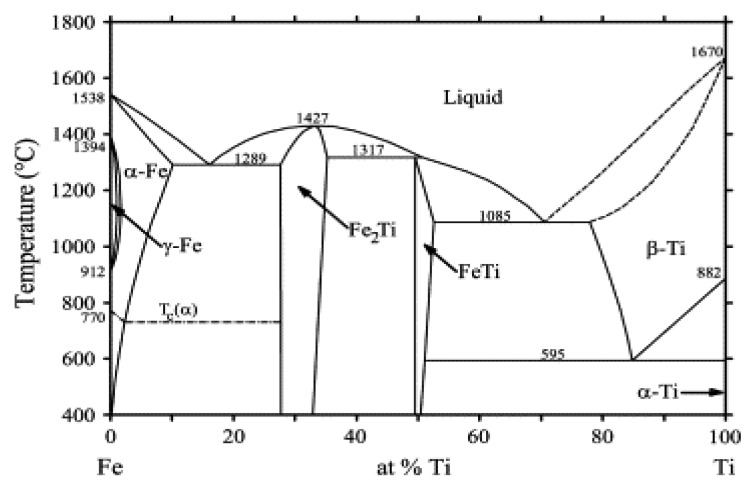
Phase diagram of Fe-Ti binary system. Reprinted from [9] with permission from Elsevier.

**Figure 2 materials-13-04747-f002:**
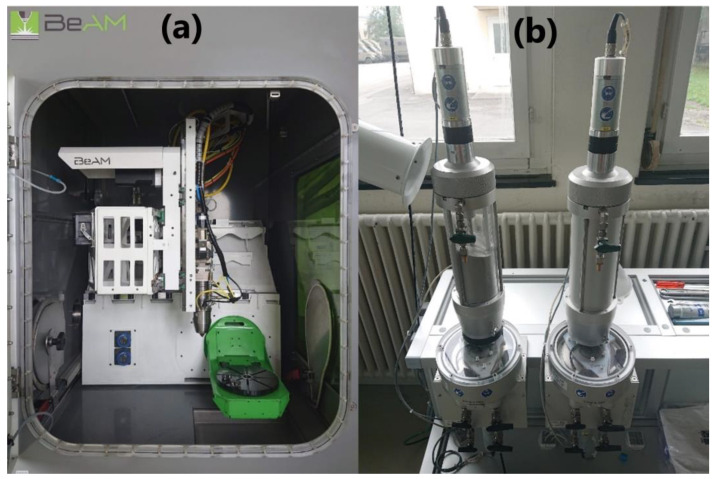
(**a**) The interior of the process chamber of the laser metal deposition (LMD) machine. (**b**) The powder feeder with two powder containers.

**Figure 3 materials-13-04747-f003:**
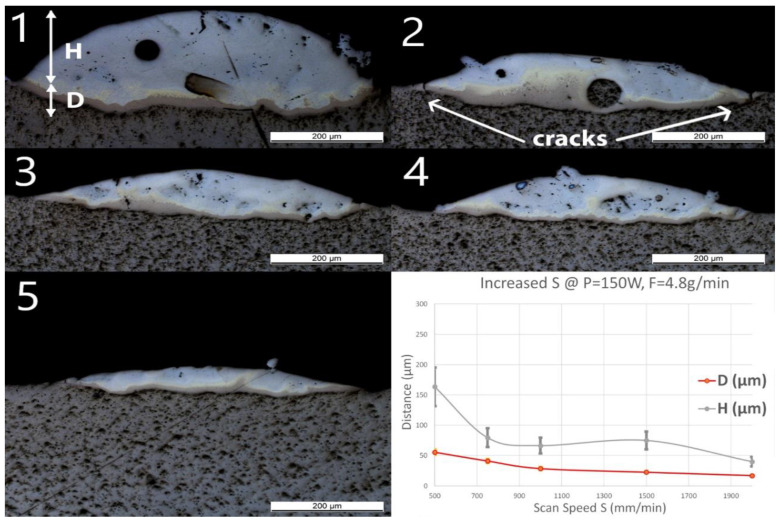
Optical microscopy (OM) images of cross-sections of samples Line_S_1 to Line_S_5 (The inserted number 1 to 5 corresponds to the sample number in Table 1). The last diagram (bottom right) shows the influence of speed (S) on penetration depth (D) and deposition height (H) of the tracks.

**Figure 4 materials-13-04747-f004:**
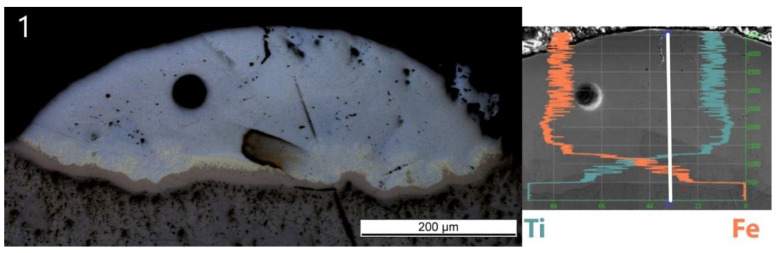
OM image of Sample Line_S_1 (left) and its energy-dispersive X-ray spectrometry (EDS) line scan result with the same scale (right). The scan was conducted along the white line. Vertical coordinate is the position on the scan line. Horizontal coordinate is the atomic concentration with 100% on the left. The cyan curve starting from 100% at the bottom (Ti substrate) is atomic concentration of Ti. The orange curve starting from 0% at the bottom is atomic concentration of Fe.

**Figure 5 materials-13-04747-f005:**
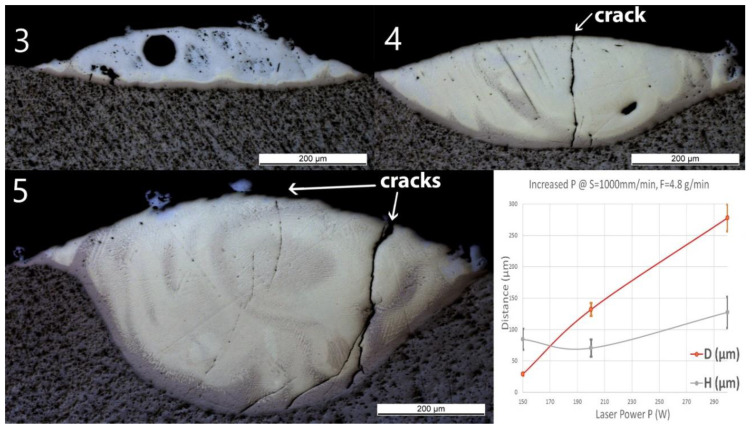
OM images of cross-sections of samples Line_P_3 to Line_P_5 (corresponding to the inserts 3 to 5 in the images above). The last diagram (bottom right) shows the influence of power (P) on penetration depth (D) and deposition height (H) of the tracks.

**Figure 6 materials-13-04747-f006:**
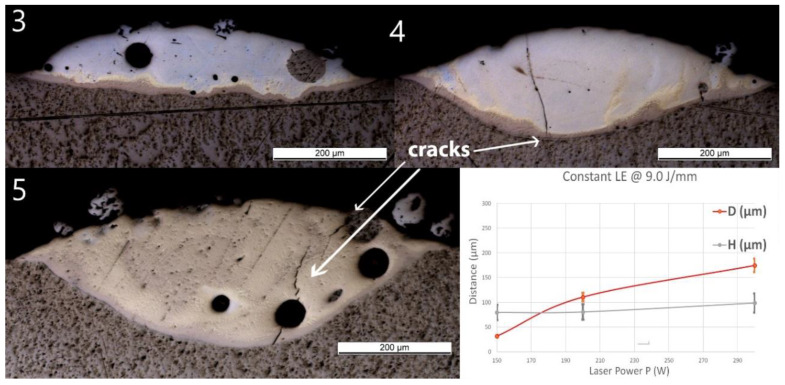
OM images of cross-sections of samples Line_LE_3 to Line_LE_5. The last diagram (bottom right) shows the influence of power (P) (together with proportionally increased speed S, maintaining a constant laser energy of LE = 9.0 J/mm) on penetration depth (D) and deposition height (H) of the tracks.

**Figure 7 materials-13-04747-f007:**
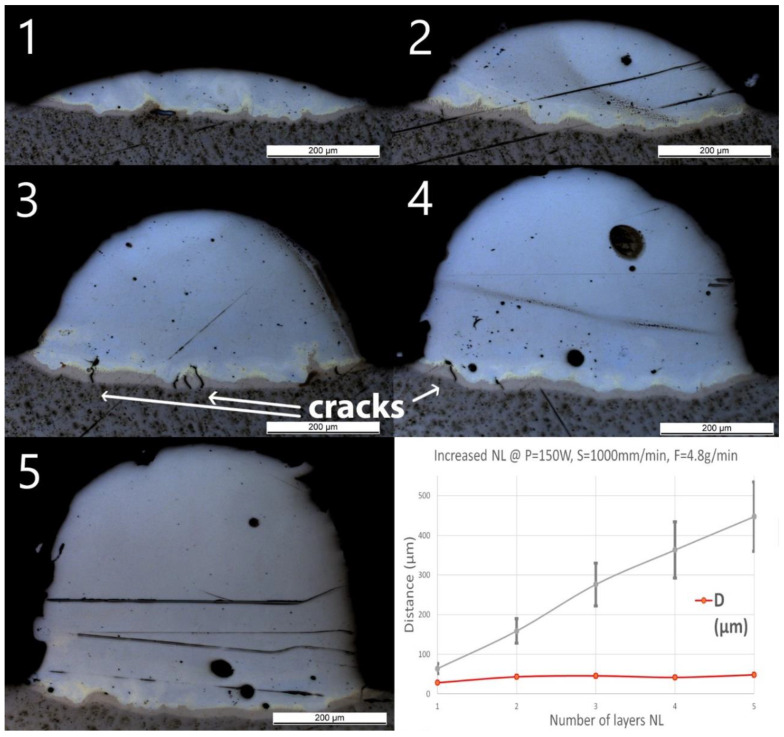
OM images of cross-sections of samples Wall_NL_1 to Wall_NL_5. The last diagram (bottom right) shows the influence of number of layers (NL) on penetration depth (D) and deposition height (H) of the tracks.

**Figure 8 materials-13-04747-f008:**
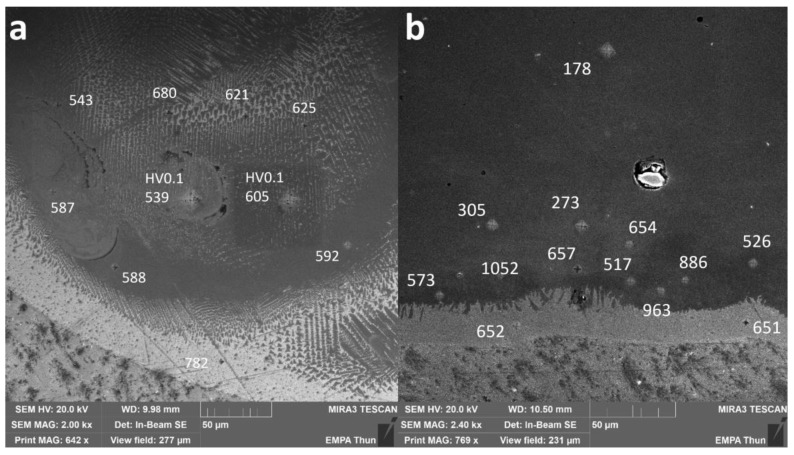
SEM images of cross-sections of (**a**) Line_P_5 and (**b**) Wall_NL_5. Numbers without specification are HV_0.01_ values. Two points were measured as HV_0.1_ to compare with results from multiple grains covered.

**Figure 9 materials-13-04747-f009:**
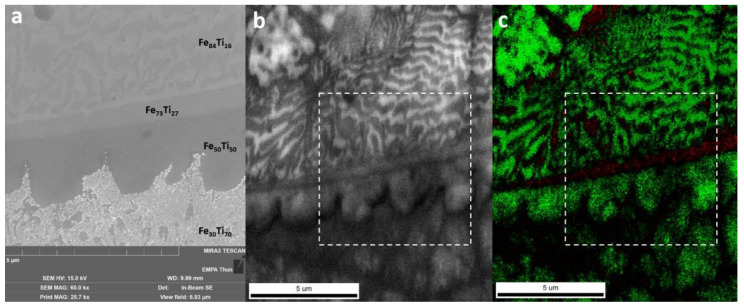
(**a**) SEM image of the upper border of the Ti-rich eutectic band in Wall_NL_5. Compositions are indicated by atomic ratios. This area is indicated by white boxes in the following electron backscatter diffraction (EBSD) maps. (**b**) EBSD image quality (IQ) map. (**c**) EBSD phase map (green for b.c.c. generic and red for h.c.p. generic) with CI (brightness indicates confidence of indexing).

**Table 1 materials-13-04747-t001:** Parameters used for all samples.

Group	No.	P (W)	S (mm/min)	F (g/min)	NL	CT (s)	LE (J/mm)
**Line_S**	1	150	500	4.8	1	/	18.0
	2	150	750	4.8	1	/	12.0
	3	150	1000	4.8	1	/	9.0
	4	150	1500	4.8	1	/	6.0
	5	150	2000	4.8	1	/	4.5
**Line_P**	1	75	1000	4.8	1	/	4.5
	2	100	1000	4.8	1	/	6.0
	3	150	1000	4.8	1	/	9.0
	4	200	1000	4.8	1	/	12.0
	5	300	1000	4.8	1	/	18.0
**Line_LE**	1	75	500	4.8	1	/	9.0
	2	100	665	4.8	1	/	9.0
	3	150	1000	4.8	1	/	9.0
	4	200	1330	4.8	1	/	9.0
	5	300	2000	4.8	1	/	9.0
**Line_F**	1	150	1000	2.4	1	/	9.0
	2	150	1000	3.6	1	/	9.0
	3	150	1000	4.8	1	/	9.0
	4	150	1000	5.8	1	/	9.0
	5	150	1000	6.8	1	/	9.0
**Wall_NL**	1	150	1000	4.8	1	3	9.0
	2	150	1000	4.8	2	3	9.0
	3	150	1000	4.8	3	3	9.0
	4	150	1000	4.8	4	3	9.0
	5	150	1000	4.8	5	3	9.0
**Wall_CT**	1	150	1000	4.8	3	8	9.0
	2	150	1000	4.8	3	5	9.0
	3	150	1000	4.8	3	3	9.0
	4	150	1000	4.8	3	2	9.0
	5	150	1000	4.8	3	1	9.0

**Table 2 materials-13-04747-t002:** Compositions obtained by EDS point analyses with corresponding microhardness and possible phases with their crystalline structure that can be found in the Fe-Ti phase diagram in Figure 1.

Composition (Atomic Ratio)	HV_0.01_	Phases
Ti	154 ± 18	α-Ti (h.c.p.)
Fe_30_Ti_70_	652 ± 1	Eutectic of β-Ti (b.c.c.) and FeTi (b.c.c.)
Fe_50_Ti_50_	539 ± 22	FeTi (b.c.c.)
Fe_73_Ti_27_	967 ± 85	Solubility limit of Fe in Fe_2_Ti (h.c.p.)
Fe_84_Ti_16_	656 ± 2	Eutectic of Fe_2_Ti (h.c.p.) and α-Fe (b.c.c.)
Fe ≥ 90%	252 ± 74	Basically α-Fe (b.c.c.)

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
