# Peer review of "Interface Control in Additive Manufacturing of Dissimilar Metals Forming Intermetallic Compounds—Fe-Ti as a Model System"

_materials, 2020, doi:10.3390/ma13214747_

Round 1

Reviewer 1 Report

The authors greatly improved the overall quality of the work by focusing the article on the deposition of Fe on Ti through LDM process and evaluating the intermetallic compounds that can form and how they affect the hardness of the joint. Now the paper is more logically arranged. The reviewer suggests accepting the paper after minor revisions:

Figure 3: the caption still refers to the previous image containing the EDS results. Please, remove the second and third sentences.

Line 175: please, use “not significant” instead of “insignificant”.

Author Response

We thanks the reviewer for his inputs and for accepting our revised manuscript. Both minor revisions have been done accordingly.  

  1. The caption of Figure 3 has been modified removing the reference to EDS. 
  2. Insignificant has been changed by not significant. 

Reviewer 2 Report

Laser metal deposition of dissimilar metals (Fe-Ti) is an industrial issue of high importance. However, some points have to be addressed before the paper can be recommended for publication:

Images of cross sections are of low quality (with many scratches). Please, improve the quality of micrographs. Instead the tiny numbers (corresponding to height or coordinates, perhaps?), schematic sketch of penetration depth and deposition depth should better be presented in images.

In fig. 3, the EDS overlays are not present (contrary to the legend).

Lines 271 and 276 - metallurgical “bounding” should probably be metallurgical “bonding” (bounding=limitation).

Cited literature could be extended.

Author Response

We thank the reviewer for the comments and tried to address all the highlighted points. 

The OM images have been improved and we removed the red lines used for measuring the thicknesses. However, we agree many scratches are still visible but that the best quality we achieved after many trials using conventional metallurgical preparation recipes. We also have to mention that we only wanted to characterize the dimensions of the layers and to have an idea about delamination, interfacial strength and porosity. Therefore, the quality was good enough. We now focus on the system Ti-Stainless steel and here, of course, we plan EBSD and more advanced techniques for the phases identification. Our metallurgical lab is then involved for the preparation. 

We have added arrows for showing the geometrical characteristics H and D on the first OM image.

The reference to EDS in caption of Figure 3 has been removed. 

"bounding" has been replaced by "bonding" lines 271 and 276 (now 272 and 277) but additionally also line 265. 

Reviewer 3 Report

Dear Authors!

I have reviewed your manuscript "Interface control in additive manufacturing of dissimilar metals forming intermetallic compounds - Fe-Ti as a model system". The subject of the research is undoubtedly interesting and worth to be studied. Unfortunately, the manuscript is not recommended for publication in its current shape. My comment are to be addressed in the revised version.

  1. Typos and errors:

Line 13 "is formed" -> "are formed"

Line 135 "the interface and the deposition" -> "the interface of the deposition"

Table 2 "Solubility limit of α-Fe in Fe2Ti IMC" -> "Solubility limit of Fe in Fe2Ti IMC".

"Nb contamination from previous experiments" - is it the issue of a single sample? Are other samples also affected? I would recommend to clean the chamber and re-do the samples with Nb contamination.    

2. Methods. 

I suggest that an etching agent was used after mechanical polishing. Please, specify this as well as the use of polarized light to get the contrast in the Figures 3-7. 

I also recommend to present SEM images at larger magnifications and (!)  apply EBSD - it is NOT fully correct to discuss intermetallic phases having chemical compositions only. 

3. Results.

I see different lines in the view-field at OM images - you mention cracks. Please, mark cracks with arrow (or other markers).

Figures 3-7 contain many straight lines. What are they? Scratches from mechanical polishing? Layer interfaces (Figure 7)?

4. Discussion. 

I would like you estimate the cooling rate. The reason is very simple - rapid cooling is well know method to change the phase composition from that of  equilibrium one. Solubility ranges are often extended. Intermetallics often solidify as amorphous phase or metastable intermetallics.  Thus, referring to phase diagram is NOT fully correct. 

EBSD may give you information on the shape of grains (dendrites ?) and phase composition in the intermediate and interface area. 

 Until you acquire these data you conclusions must be revised in much more accurate manner not referring to binary phase diagram.

Regaras ... 

Author Response

We thank the reviewer for finding our research interesting and for the improvement suggestions. Please find below our comments to your raised points: 

The mentioned typos and errors have all been fixed.

We were also disappointed to find one remaining Nb particles from previous experiments performed with the same DED facility. This machine is used for many projects on different material systems and cross-contamination is difficult to avoid. However these niobium particles were only found in two samples and due to the higher melting point of Nb compared to Ti and Fe, these particles were not melted and did not contribute to any phase modification, at least at the macroscale observations made in this study.

We did not etch the cross sections, we only polished with an OPS solution containing 10% H2O2.

The cracks have been highlighted with arrows as recommended (see Fig. 3-2). The straight lines observed on the cross sections are from polishing. A sentence has been added in the manuscript (Line 142-144).

We could not distinguish the interface between two subsequent deposited layers. A sentence has been added to the manuscript line 222-223.

The present study is a preliminary study aiming at understanding the interfacial transitions between Ti and Fe. Fe is not the interesting material for applications but we used it as a model for stainless steel as the critical intermetallics reported in the SS-Ti system are FeTi and Fe2Ti. We are now working on this more complex material system and advanced characterizations requiring a much better surface preparation are performed, like for instance EBSD. 

We fully agree with your remark concerning the non-equilibrium state of the process and the difficulty to refer to phase diagrams. In the present case, we did not perform any measurement of the cooling rate but it is well reported in the literature. The only information we have to try to identify the phases is however these phase diagrams calculated or measured under thermal equilibrium. We are aware that some metastable phases may form but, in this study, we could find almost all the phases reported at equilibrium . We then added sentences in the manuscript for highlighting the non-equilibrium of the laser process (line 42-43, 310 and in the conclusion line 322). 

Round 2

Reviewer 2 Report

The paper is OK now.

Author Response

Thank you for your support.

Reviewer 3 Report

Dear Authors! I appreciate your efforts to slightly revise your submission addressing to some of my comments. On the other hand, the principal point - namely, referring to phase diagram (intrinsically describing  equilibrium state) in the situation of non-equilibrium process - has not been satisfactory answered. I general, the chemical composition mapping is NOT good enough to speculate about phases, until the cooling rates will be estimated/assessed or found in the literature. I understand, that EBSD is not available at the moment, but literature search is always with you, right? If you demonstrate that cooling rates characteristic for your process do NOT affect phase composition - this would be a stronger argument. 

Regards ...

Author Response

Dear Reviewer,

you are completely right with the non-equilibrium of the process and we fully agree. The cooling rates encountered in DED process are one or two orders of magnitude lower than those calculated or measured for SLM processes due to much lower velocities. Anyway the high cooling rates may induce some changes of the phase borders, the solubility limit normally increases, and finally some metastable phases could also occur. 

We performed XRD measurements but due to the small volumes of the different phases, no information could be extract from diagrams and thus this is not presented in the paper. We then had to perform EBSD on one interfacial transition between Ti and Fe and we implemented these results into the paper. However as you know many phases in this material system have the same crystalline structure hcp or bcc and therefore the assignment of the phases is not easy however we could find a good correlation between the expected phases based on EDS compositions and the crystalline indexation of EBSD. You find the results in Figure 9 besides the higher magnification SEM image and sentences have been added in the manuscript. The EBSD method has been also described in the part method. 

We hope therewith to have convince you about the phases forming at the interface. 

Thank you again for the improvement suggestions. 

This manuscript is a resubmission of an earlier submission. The following is a list of the peer review reports and author responses from that submission.

Round 1

Reviewer 1 Report

Authors focused on LMD of dissimilar metals, which is an interesting subject and fits the scope of Materials journal. However, I was quite confused by the declared aims of the paper and experiments performed (and presented).

The motivation (if I well understood) is the joining of Ti and SS. Then studying Fe-Ti system could be a good idea. Why is, indeed, OES applied to SS-Ti LMD process and not also to Fe-Ti? How (if it possible) can the results on Fe-Ti system be transferred to SS-Ti system? This is not adequately explained (or discussed – the discussion is by the way rather hazy). Without clear link between Fe-Ti and SS-Ti systems, the paper is completely loosing sense.

In my opinion, the paper should be rewritten in more logically arranged form, otherwise it cannot be recommended for publication.

Other remarks.

Consider using simple past instead present perfect tense.

The figures with EDS line insets are very hard (impossible) to read (see). Consider other type of presentation.

Avoid excessive using of “However”.

Author Response

We thank the reviewer for his comments, we changed the manuscript accordingly and we hope to answer the remarks positively point-by-point below:

We understand that the OES part performed on stainless steel instead of iron induces some confusion. We thus remove the OES part, reducing then the confusion between SS and Fe and hope therewith that the paper is presented in a more logical way.  

The English has been checked.

The EDS line scans are indeed not readable especially for thin deposition layers. What is important is the trend and the different measured compositions are reported then in the manuscript. We think that having all the information of one experimental series on one figure is showing better the differences. This presentation has been kept for all the series. 

We indeed performed OES measurements mostly on the Ti-stainless steel system and not really on the Ti-Fe. We thus agree that the OES part is not well integrated in this paper and we remove it. The main message out of these analyses is that we can detect (in-situ and in-real time) if the first metal deposited (or the substrate) is still present in the melt pool. It should help in the future to adjust the parameters for controlling better the interface and then for optimizing the deposition parameters of the second metal. This will be presented in a future paper.

We are aware that investigating the Fe-Ti material system whereas applications in medical devices and implants rather involve stainless steel and titanium may be confusing. We wanted to avoid increasing the complexity of the interfacial zone by limiting the number of phases able to form like σFeCr, αCr2Ti, βCr2Ti, γCr2Ti and Ti5Cr8Fe16and thus selected only Fe that may build the most critical intermetallics with Ti. We emphasize again the justification of only using Fe instead of stainless steel in the introduction. 

Reviewer 2 Report

The authors report a well written manuscript aimed at demonstrating the technological feasibility in joining stainless steel and titanium alloys through laser metal deposition. They highlight the presence of a thin brittle intermetallic zone which does not get wider for further depositions and which is aimed at tayloring the interface between the two different metals where cracks and delamination would occur. However, they do not report on the strength of such joint, but state that this aspect is under investigation. Moreover, they are missing on experimental details in terms of deposition of Ti on SS. The reviewer suggests to reconsider the article after major revisions.

Line 25: What does "SS" stand for? No definition is reported in the abstract.

Line 92: the authors state that they perform the deposition of Ti on SS and SS on Ti. However, in the "Materials and Metods" no information is given about the Ti powder and Fe substrate adopted, neither about the process parameters. In addition, the "Results" and the "Discussion" sections seem to be only devoted to the deposition of SS on Ti substrate, while merely Figure 8 shows some results about the deposition of Ti on SS. Please, provide more details about the deposition of Ti on SS both in terms of materials and results.

Line 107: please, provide a figure with the experimental setup.

Table 1: is the first line of "WALL_NL" correctly highlighted?

Line 142: please, control the brackets.

Figures 2-5: the font size of the diagrams on the right side of the sub-figures are not readable. Please, avoid overlapping images and scale lengths.

References: the authors self-cite their works for 40% of the references.

Author Response

We thanks the reviewer for his improvement suggestions. We accordingly changed the manuscript and address the comments point-by-point below:

"SS" stands for "stainless steel". As the abstract has been revised, SS first appears in the introduction with its signification.

As reported also by another reviewer the OES study on Ti-SS instead of Ti-Fe may makes the paper unclear. We agree and we remove this part completely just letting a short remark in the discussion about the capability of OES to detect the presence of the bottom metal. This is the topic of one current study not published yet on optimizing the interface and the process efficiency for Ti-SS.

Due to the removal of the OES part on the SS-Ti material system, the results and discussion parts are more focused on the investigated primary system Fe-Ti. We were not able to find published work on Fe-Ti and therefore we discussed our results based on the previous studies performed on SS-Ti. 

A figure has been added showing the experimental setup. 

All the lines of the table highlighted with light grey are single tracks printed with the standard parameters. Those with dark grey are single walls (three layers) with the same standard parameters. We wanted to investigate the reproducibility of the process and thus repeat deposition  conditions many times. 

We agree that these figures may be overloaded but we find that bringing all information of one experimental series on only one figure is beneficial. We also keep the same presentation of the results for all the different series. However we agree that the readability is not optimum and we tried to improve it.

We did not report on mechanical strength of the interfacial zone. Only microhardness measurements were performed but we never observed delamination or long cracks along the interface. It is in contradiction to previous reported studies on stainless steel-titanium. We think that the main difference is the volume of the melt pool as the authors who observed delamination during cooling were using much higher energy inputs thus promoting the melting of both metals and consequently the formation of these two brittle intermetallic phases FeTi and Fe2Ti. This is addressed in the discussion. 

Most of the self-citations were coming from the OES part. As we removed this part, the number of self-citations dramatically decreased.

Round 2

Reviewer 1 Report

The questions and remarks made in the first round of review were not satisfactorily answered.

I insist that without clear link between Fe-Ti and SS-Ti systems, the paper is completely loosing sense and the link is still missing in the revised version.

The OES part was removed from results, nevertheless, it is still in the discussion. Authors claim that OES can detect if the first metal deposited (or the substrate) is still present in the melt pool, and what? This is not further developed, by the way, can the change of composition from Fe80Ti20 to Fe90Ti10 within the first 100 μm be directly detected by OES (how precisely etc.)? This is confusing.

English was not improved.

The figures with EDS line insets are not readable and the authors refuse to improve them (even if they agree on this fact !).

I do not recommend the paper for publication in the present state.

Reviewer 2 Report

The authors greatly improved the overall quality of the work by focusing the investigation on the deposition of Fe powder on Ti substrate. The reviewer suggests to accept the article after minor revisions:

Lines 312-325: in this paragraph the authors describe the optical emission spectra of single SS line on Ti. Is SS correct? The reviewer recommends to describe this part with Fe instead of SS to be in line with the text. Or did the authors perform the deposition of SS on Ti? Why did they use different process parameters from those adopted in the experimental investigation? Please, clarify.